# Aspartic Acid Isomerization Characterized by High Definition Mass Spectrometry Significantly Alters the Bioactivity of a Novel Toxin from *Poecilotheria*

**DOI:** 10.3390/toxins12040207

**Published:** 2020-03-25

**Authors:** Stephen R. Johnson, Hillary G. Rikli

**Affiliations:** 1Carbon Dynamics Institute LLC, Sherman, IL 62684, USA; 2Chemistry Department, University of Illinois Springfield, Springfield, IL 62703, USA; 3College of Liberal Arts & Sciences, University of Illinois Springfield, Springfield, IL 62703, USA; hrikl2@uis.edu

**Keywords:** Voltage-gated sodium channel, venom, *Poecilotheria*, poecilotheriatoxin, deamidation, isomerization, high definition mass spectrometry, ion mobility, supplemental activation

## Abstract

Research in toxinology has created a pharmacological paradox. With an estimated 220,000 venomous animals worldwide, the study of peptidyl toxins provides a vast number of effector molecules. However, due to the complexity of the protein-protein interactions, there are fewer than ten venom-derived molecules on the market. Structural characterization and identification of post-translational modifications are essential to develop biological lead structures into pharmaceuticals. Utilizing advancements in mass spectrometry, we have created a high definition approach that fuses conventional high-resolution MS-MS with ion mobility spectrometry (HDMS^E^) to elucidate these primary structure characteristics. We investigated venom from ten species of “tiger” spider (Genus: *Poecilotheria*) and discovered they contain isobaric conformers originating from non-enzymatic *Asp* isomerization. One conformer pair conserved in five of ten species examined, denominated PcaTX-1a and PcaTX-1b, was found to be a 36-residue peptide with a cysteine knot, an amidated C-terminus, and *isoAsp*33*Asp* substitution. Although the isomerization of *Asp* has been implicated in many pathologies, this is the first characterization of *Asp* isomerization in a toxin and demonstrates the isomerized product’s diminished physiological effects. This study establishes the value of a HDMS^E^ approach to toxin screening and characterization.

## 1. Introduction

In the 1970s and 1980s William Catterall and colleagues discovered that isolated scorpion venom components bound with high affinity to ion channels. Their investigations led to discovery of the voltage-gated sodium channel (VGSC) complex [1,2]. From these early studies, we now know that there are nine isoforms of VGSCs (denoted Na_v_1.1–1.9) that initiate and propagate the rising phase of an action potential [3]. Since then, VGSCs have been studied intensively to further elucidate their functional roles in normal physiology and pathologies, making them an attractive target for therapeutic intervention and drug development [4,5,6,7,8]. More specifically, investigations of VGSC isoforms expressed in the peripheral nervous system (Na_v_1.7–1.9) are ongoing due to their involvement in pathological pain [9,10,11,12,13]. Gain of function mutations in VGSCs have been shown to participate in several chronic neuropathies: inherited erythromelalgia, paroxysmal extreme pain disorder, and small fiber neuropathy [14,15,16]. In an odd coincidental reality, a loss of function mutation, known as congenital insensitivity to pain, ameliorates the perception of pain completely [17,18], making the Na_v_1.7 a target for non-opioid analgesics and a critical focus of research [19].

Current studies are focused on developing a stronger understanding of how toxins modulate the kinetic behavior of ion channels [20,21]. Nature has provided a bountiful array of effector molecules having myriad chemistries that have been observed, investigated, and characterized to explain these consequential interactions [22,23,24,25]. These studies have illuminated the modulation of VGSCs [9,26]. There are six different receptor sites (site 1–6) on VGSCs where binding by a neurotoxin has shown similar effects that may be useful in explaining their ability to alter the development or maintenance of action potentials [27,28]. Perhaps the most useful aspect of developing a ligand library is the opportunity to identify conserved features in toxin structures that display common modulatory effects for use in future drug design considerations [29,30,31].

The challenge in building this library is the ability to visualize the everchanging temporal and spatial expression of biomolecules and couple this to an essential phenome. While the primary structure of a protein is important, advanced techniques utilized in current studies have allowed us to move beyond the strict sequence of amino acids to develop a profile of conserved features in these toxin modulators. While sequencing amino acids does not necessarily predict activity, we can look at PTMs for better understanding of their modulatory effects. The significance of these structures is undeniable as they can be found on greater than 40% of all proteins in the UnitProt Database [32,33]. These modifications exert their actions by modulating protein-protein interactions (PPIs) [33]. An early discovery that probed a requisite PTM in activity studies was the cysteine knot motif with specific bridging essential for toxin activity [34,35,36]. Although cysteine knots have other proteolytic functions in vertebrates (blood clotting, digestion, and fibrinolysis), they are a divergent feature found in many invertebrate venoms to protect the peptide from serine protease digestion [37]. Another classic example of this proteomic strategy is amidation of the C-terminus by the sequential action of peptidyl glycine α-hydroxylating monooxygenase (PHM) and peptidyl-α-hydroxyglycine α-amidating lyase (PAL) found in primitive species or a bifunctional protein peptidyl α-amidating monooxygenase (PAM) found in mammals [38]. This C-terminus modification renders the peptide less sensitive to proteolytic degradation and may increase the affinity of a peptide to its receptor [39,40,41,42]. 

One PTM often seen in proteomic investigations is deamidation of *Asn* and isomerization of *Asp* [43]. Deamidation and isomerization result from spontaneous non-enzymatic formation of a succinimide intermediate due to the nucleophilic attack by the nitrogen atom in a succeeding residue on the carbonyl group of the side chain of *Asn* or *Asp* [44] (Figure 1). However, spontaneous degradation does not always occur. Oftentimes, the aspartyl and asparaginyl residues are embedded in conformations where the peptide nitrogen atom cannot approach the side chain carbonyl carbon to form a succinimide ring [45]. Although the presence of this beta amino acid, i.e., *isoAsp*, can significantly alter the structure from their precursor analog, they are difficult to detect in normal proteomic workflows. A deeper understanding of this PTM may provide new insights of toxin modulation that adds to the library of higher order structure-activity relationships (SARs). 

For the purpose of this study, we employed a comprehensive high definition mass spectrometry (HDMS^E^) approach to investigate the venom from ten species of an arboreal tarantula commonly known as the “tiger” or “ornamental” spider (Genus *Poecilotheria*): *P. striata, P. regalis, P. rufilata, P. formosa, P. metallica, P. miranda, P. tigrinawesseli, P. fasciata, P. ornata, P. smithi,* and *P. vittata.* While there have been studies of whole venom from this genus of aggressive tarantula [46,47,48,49], our laboratory is the first to characterize a complete primary structure of a poecilotheriatoxin as well as evaluate the impact isomerization has on a toxin’s physiological effect(s). Using a combination of digestion strategies, fragmentation techniques, and tandem high-resolution ion mobility spectrometry (IMS—interchangeable in this manuscript as high definition mass spectrometry HDMS^E^), potential candidates for one non-enzymatic PTM was identified and confirmed, i.e., isomerization of *Asp*. While all the above strategies were used for complete characterization of a peptide, electron transfer dissociation (ETD) played the most important role in confirming the isomerization of *Asp.* In this study, we demonstrated our biochemical strategy for the characterization of a novel VGSC modulator (β/δ/μ-theraphotoxin-Pv1 or β/δ/μ-TRTX-Pv1 in the rational nomenclature [50]) from the venom of *Poecilotheria vittata*. This poecilotheriatoxin (PcaTX-1) is a 36-residue peptide with three disulfide bridges and an amidated C-terminus with the primary structure:



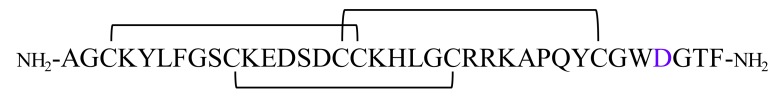



The *Asp*^33^ (shown in purple designates the presence of an *isoAsp* or an *Asp*) was the only *Asp* residue found susceptible to isomerization and displayed significantly different modulation from the native isoform. *Asp*^13^ and *Asp*^15^ resisted isomerization under basic conditions. Taken together, these studies reveal the importance of primary structure, PTMs, and their effect on the VGSC Na_v_1.7 modulation.

## 2. Results

### 2.1. Toxin Selection

We used an iterative series of analytical processes to identify potential peptide modulators for neuronal activity and subsequently identified their primary structure, including PTMs, that may alter their efficacy and potency. In the initial phase of toxin selection, venom from *Poecilotheria* were collected and screened by HDMS^E^. The venom was a complex mixture of molecules with a high percentage of proteinaceous materials ranging from 3–6 kDa with an average venom yield from 50 collections (five from each of the ten species) of ~148 ± 12 μg of total protein per μL of venom as determined by conventional Bradford assay (data not shown). An analytical screening of each venom was performed for preliminary compound identification and any PTMs, e.g., isobaric conformations. 100 μg of lyophilized venom from each species of *Poecilotheria* were suspended in 1 mL of the initial mobile phase and a 1 μL aliquot injection was made into an UPLC coupled to a HDMS. All initial screenings were acquired in electrospray positive ion mode (ESI+) HDMS^E^ providing conventional UPLC separation and high-resolution MS^E^ spectra with an additional dimension of separation based on molecular size and shape: IMS. 

The total ion chromatograms (TICs) of venom for each species of *Poecilotheria* demonstrated a variety of low molecular mass peptides with a broad range of partition indices and very little conservation between species (Figure 2).

After developing a mass list of peptides for each species based on retention time (*t*_R_), several potential conformer groups were observed in each species. In order to demonstrate an example of a representative conformer group in each venom, an extracted ion chromatogram (EIC) was filled (purple) and overlaid on each TIC for clarity (Figure 2). One conserved conformer group, identified as Poecilotheriatoxin-1 (PcaTX-1), was found in five of the ten species examined (EICs—purple filled) and had an average molecular mass of 4027.6 Da (Figure 3). We use the genus name from binominal nomenclature for the identifier, because in on our investigations we find there are several toxins conserved among species. Historically, many toxins are named from the species in which they are first characterized, or more recently, the family from which the species is contained. A second reason we keep the nomenclature simple at this point in the investigation and do not assign an activity identifier is that we do not always know the physiological role a toxin may serve on all potential targets. Oftentimes this can be promiscuous or dependent on the species affected so naming the peptide in that fashion can be misleading.

In order to calculate the molecular mass, we deconvoluted the charge envelope of each peptide using high-resolution mass spectrometry MS scans (“MS” in HDMS^E^). The TIC (250–2000 *m/z*) for *P. vittata* was provided as a reference to show an EIC (806–807 *m/z*—filled purple) that demonstrated the presence of isobaric conformers (Figure 3A). Although there were seven peaks observed in the EIC, only three were isobaric conformers with the same monoisotopic mass: the peaks at *t*_R_ = 4.83, 5.02, and 5.26 min, respectively. While the conformer peptide at *t*_R_ = 5.02 min is the subject of a different manuscript to be published in the future, the two most abundant peptides, i.e., *t*_R_ 4.83 min and *t*_R_ 5.26 min, were chosen for further characterization and hereafter referred to as PcaTX-1a (data displayed in red) and PcaTX-1b (data displayed in blue). The molecular monoisotopic masses of PcaTX-1a and PcaTX-1b were determined to be 4024.710 Da and 4024.709 Da, respectively (Figure 3B,C). A high-resolution MS-MS acquisition with a collision energy ramp (25–55 V) was performed for ions above a set threshold (“^E^” in HDMS^E^) as a data independent acquisition (DIA). The MS^E^ data revealed similar charge permutation reactions for both conformers typical of compact peptides stabilized by disulfide bridging suggesting the presence of a cysteine knot (data not shown). 

To provide an orthogonal view of the mass spectra based on the mobility of charged gas-phase ions through a carrier gas in an electric field (“HD” in HDMS^E^), a traveling wave IMS (TWIMS) technique was employed. This technique applied a sequence of symmetric potential waves through the drift tube that propelled ions along with velocity dependent on *K* (where *K* is the mobility coefficient in the expression *v* = *KE*) [51]. Ions in a traveling wave ion guide are separated based on their collision cross section (CCS, Ω) which is a function of their average arrival time distributions (ATDs). This ion mobility approach has the unique capability to separate ions based on charge, size, and shape. Thereby, smaller ions exit the ion guide faster than larger ones. Similarly, molecules with different charge states migrate differently: the larger the charge state, the faster their mobility (Figure 4A). To determine if there were differences in the overall shape of the isomeric peptides in the gas-phase, we relied on chromatography and ion mobility to further elucidate possible candidates. By selecting different charge states of a molecular mass, ions with different conformations can be separated. A representative extracted ion cluster, i.e., [M + 5H]^5+^, is provided (Figure 4B). Since all these ions have the same mass as well as the same shape, we were unable to identify any other possible conformers at the PcaTX-1 *t*_R_’s, but we were able to observe that all the major charge states did not significantly differ in mobility (Figure 4C,D). Although we postulated a change in shape may have explained the differences observed in both chromatographic retention (Figure 3) and physiological effects (Figure 5), we concluded a more subtle feature must be responsible. We were able to confirm that there were no co-eluting ions suggesting that the chromatography had adequately resolved this group. 

### 2.2. Effect of PcaTX-1 Conformers on Na_v_1.7 Currents

Previous research has shown toxins derived from Theraphosidae venoms contain a rich assortment of potential VGSC modulators [20,24]. In order to isolate and purify a candidate, ~5 mg of freeze-dried venom from a species was subjected to size exclusion chromatography (SEC), followed by large volume reverse-phase chromatography (RPC) using a semipreparative column. One important note is there were other “isomeric” candidates observed (Figure 3A EIC) but were later determined to be peptides with similar masses, isomers with different disulfide bridge configurations, or isomers with different primary structures. 

To probe for ion channel modulation, CAD cells that dominantly express Na_v_1.7 channels were clamped in a whole-cell configuration and were allowed to equilibrate for 5 min prior to any analyses [52]. Purified PcaTX-1a and PcaTX-1b (diluted in extracellular solution at a concentration of 500 μg/L (~125 nM) immediately before analyses) were perfused at 0.5 mL/min and post drug analyses were not recorded until a maximum and equilibrated effect were observed, to alleviate differences that may occur in binding onset of the conformers to the ion channels. Currents were induced by 50 ms depolarizing steps in 5 mV increments from −75 to +75 mV from a holding potential (*V*_H_) of −100 mV. Representative current traces for the control, PcaTX-1a, and PcaTX-1b are provided to demonstrate the toxins’ overall effects on Na_v_1.7 currents (Figure 5A). Both peptides dramatically slowed inactivation inducing a prolonged opening of the VGSC, suggesting a site 3 interaction. In addition, PcaTX-1b altered the gating properties of Na_v_1.7 by reducing the amplitude of the peak Na^+^ current and caused a shift of voltage dependent activation, suggesting a site 4 interaction. Taken together, these data suggest PcaTX-1 conformers are VGSC modulators, with dual binding sites for the PcaTX-1b activity. 

To further characterize the modulatory effects of PcaTX-1a to PcaTX-1b on the activation kinetics of Na_v_1.7, a normalized current-voltage (*I/V*) plot was generated and fitted (Figure 5B). The half-maximal voltage (*I V*_½_) of -32.9 ± 0.889 mV from the control was not significantly different from −33.6 ± 0.683 mV for PcaTX-1a (Table 1). However, the *I V*_½_ of −14.5 ± 0.592 mV from PcaTX-1b was significantly different than the control and PcaTX-1a. No change in reversal potential (*V*_rev_) was observed in either treatment. To examine the voltage dependence of activation, the conductance-voltage (*G/V*) curves were plotted and fitted with a Boltzmann distribution (Figure 5C). The half-maximal activation of current (*G V*_½_) for the control was −28.6 ± 0.861 mV and did not significantly shift in the presence of PcaTX-1a with a *G V*_½_ of −28.4 ± 0.768 mV. However, the *G V*_½_ of −6.34 ± 0.490 mV for PcaTX-1b was significantly different than the control and PcaTX-1a. There was no difference in the slope factors (*G k*) for the control and the PcaTX-1a treatment, however, PcaTX-1b significantly shifted the slope as compared to both the control and PcaTX-1a. Though dose dependency was not performed at this stage of characterization, we assumed that PcaTX-1b is more potent and/or more efficacious than PcaTX-1a. Taken together, these data suggest PcaTX-1b altered the voltage dependence of activation by shifting to more depolarized potentials. This also suggested that the modulation seen by both peptides did not change the ion selectivity of the channels [53].

To further characterize the modulatory effects of PcaTX-1a and PcaTX-1b on the inactivation kinetics of Na_v_1.7, the time constants of decay for open-state fast inactivation (τ_F_) were determined. The τ_F_ for both PcaTX-1a and PcaTX-1b were plotted from −30 to +30 mV and compared to the control (Figure 5D). Both peptides slowed the τ_F_ with ~10 to 6-fold increase and were significantly different than the control. There was a significant difference between the τ_F_’s for PcaTX-1a and PcaTx-1b for the first four voltage steps. To examine the modulation of steady-state fast inactivation (*h_∞_*), a standard two-pulse protocol was performed. With a *V*_H_ of -100 mV, conditioning pre-pulses ranging from −130 to +5 mV in 10 mV steps for 50 ms, were delivered, and currents were measured at a depolarization step to 0 mV for a 20 ms duration (test pulse). It is important to note the significance of this protocol in that one, the pre-pulse potential is only held for 50 ms and two, there is no break to *V*_H_ in between the pre-pulse and test pulses (hyperpolarizing gap). It has been documented that “with longer durations of pre-pulses in the steady-state fast inactivation protocol, the ratio of slow inactivated channels increases from ~5% (100 ms pre-pulse) to ~40% (2 s pre-pulse).” [54]. Also, the hyperpolarizing gap is not included as >95% of channels recover from the fast-inactivated state within this period, and the purpose was to only look at how the peptides modulated the properties of fast inactivation [54]. Steady-state slow inactivation modulatory effects will be explored in future work. Normalized to the maximum, currents were plotted against the change in pre-pulse potentials, and the curves were fitted to a Boltzmann distribution (Figure 5E). The voltage dependence of steady-state fast inactivation (*h_∞_ V*_½_) of −58.6 ± 1.18 mV for the control was not significantly different than −54.1 ± 0.889 mV for PcaTX-1b but was significantly different than −48.6 ± 1.16 mV for PcaTX-1a. There was no significant difference in the slope factors (*h_∞_ k*). Taken together, these data suggest both PcaTX-1a and PcaTX-1b changed the kinetics of fast inactivation but showed different effects on the voltage dependency of fast inactivation. 

Our preliminary conclusions suggested the fractional block seen by the application of PcaTX-1b (Figure 5A,B) was a result of the depolarizing shift in activation and its concomitant reduction in ion driving force. After washing the PcaTX-1b, the depolarizing shift in activation and fractional block diminished while the inactivation effects remained and showed very similar effects seen for PcaTX-1a (data not shown). These data strongly suggest the PcaTX-1b shares the same binding site as PcaTX-1a, but also binds to a second site with less affinity. The data further suggests that PcaTX-1a may be less potent and/or less efficacious than PcaTX-1b in modulating the voltage dependency of activation by shifting the *G V*_½_ to more depolarized potentials but may be slightly more efficacious in modulating the voltage dependency of fast inactivation by shifting the *h_∞_ V*_½_ to more depolarized potentials. All these electrophysiology results taken together suggest there is a structural difference between PcaTX-1a and PcaTX-1b, despite their same charge, size, and shape.

### 2.3. Preliminary de novo Sequencing of PcaTX-1a and PcaTX-1b by CID

In order to determine the primary structure of PcaTX-1a and PcaTX-1b, 1 μg aliquots of purified toxins were digested with multiple enzyme treatments: trypsin, chymotrypsin, and Asp-N. Subsequently, 2 μL of each digest were subjected to HDMS^E^ in a series of collision induced dissociation (CID) and electron transfer dissociation (ETD) data dependent acquisitions. CID was performed on the [M + 2H]^2+^ ion above a set threshold. CID experiments were performed using an electric potential to accelerate peptides/peptide fragment ions into a neutral gas that disrupts bonds to provide cleavage at the carbonyl-N backbone, resulting in Biemann fragments referred to as *b* and *y* ions [55,56,57]. The preliminary amino acid sequence for both conformers were established by the de novo analysis of resultant data. Representative TICs of the tryptic digests of PcaTX-1a and PcaTX-1b are provided in Figure 6A to illustrate the differences seen in the elution of some digestion peptides. This differential elution of the tryptic peptides (Figure 6A—purple inset) suggested that the difference in the toxins could be determined by closer examination of these conformers, i.e., the tryptic peptides at *t*_R_ 4.21 min (from PcaTX-1a) and 4.34 min (from PcaTX-1b), respectively.

A preliminary sequence for each digestion product conformer was determined with 100% coverage, i.e., complete *b* and *y* ion assignments (Figure 6B). For purpose of clarity, the *b* ion series is shown for the early eluting tryptic peptide at *t*_R_ 4.21 min (Figure 6B top panel—red spectrum) and the *y* ion series is shown for the later eluting tryptic peptide at *t*_R_ 4.34 min (Figure 6B bottom panel—blue spectrum). Since these peptides were products of trypsin digestion, we examined the low mass range for a 147.1133 and 175.1195 *m/z* as an indication of a C-terminus *Lys* or *Arg*, respectively. We did not see either fragment, but we did locate a 165.1028 *m/z* in all examined CID spectra with this fragment, which is the mass defect seen when a *Phe* is amidated (ΔM = 0.9840 from an expected C-terminus 166.0868 *m/z* for *Phe*). Therefore, both peptides had the same preliminary sequence of NH_2_-KAPQYCGWDGTF-NH_2_. Other differences in the TICs were also observed, especially in the region around *t*_R_ 3.85 min and *t*_R_ 4.57 min. These fragments were analyzed and found to be alternate cleavages of the sequence region described above (data not shown). To clarify, the tryptic peptide at *t*_R_ 4.66 min from PcaTX-1a (red) and the tryptic peptide at *t*_R_ 4.80 min from PcaTX-1b (blue) both have the sequence NH_2_-APQYCGWDGTF-NH_2_.

An alternate fragmentation that can be seen in hybrid Q-ToF instrumentation and improves the confidence of peptide assignments is the production of largely overlooked “immonium” ions [58]. The immonium ion has the general structure RCH = NH_2_^+^, where R is the residue side chain and a −27 Da shift from the residue mass. The ion can be formed from a series of cleavages during CID or by direct fragmentation of an N-terminal residue. The low mass range of the tryptic peptide fragments (50–180 *m/z*) was examined, and nine residues were confirmed in these low energy collisions (Figure 6C). Since a mass threshold of 50 *m/z* was used for these experiments, the only undetected immonium ions were for *Gly* and *Ala* with a 30.03 and 44.05 *m/z*, respectively. A few more items of interest are the low abundance of *Thr* and the high threshold of *Phe*. Taken together, these data further confirm their location in the preliminary C-terminus peptide sequence provided above [58]. 

Although CID is not capable of providing discrimination of an *isoAsp* and *Asp*, the ion ratios of *b*_8_ and *b*_9_ provide a clue (Figure 6D). The differential fragmentation around the *Asp* suggested an alternate cleavage that may be due to the presence of a difference in the peptide backbone. Other digestion peptides from PcaTX-1a and PcaTX-1b determined to have *Asp* did not show these abundance ratio differences unless they were at this locus of the sequence and were always accompanied by a change in chromatographic retention. Taken together, these data suggest that this *Asp* residue located near the C-terminus was either an *isoAsp* or *Asp*. 

### 2.4. Confirmatory de novo Sequencing of PcaTX-1a and PcaTX-1b by ETD

During the data dependent assays performed in Section 2.3, ETD was coupled with CID to confirm the preliminary primary structure of the tryptic peptides on any [M + 3H]^3+^ ions that rose above set threshold; an additional scan for [M + 4H]^4+^ was added if a fragment was large enough to display that charge state, e.g., chymotrypsin digestion products. Since many PTMs, e.g., phosphorylation or glycosylation, are eliminated as a “neutral loss,” making primary structure work difficult in low energy environment (<100 eV) of CID, ETD is employed to leave labile PTMs intact with positional diagnostics [59]. ETD provided a tremendous value in confirming de novo interpretations of CID spectra and more importantly allowed for the distinction between *isoAsp* and *Asp*. ETD utilized an electron transfer in gas-phase ion-ion chemistry, causing rearrangement with cleavages of the N-C_α_ backbone referred to as *c* and *z* ions [60]. While the fragment ratios seen in CID differ suggesting structural discontinuity between the peptides, ETD cleaves the isomers of *Asp* producing “reporter” ions in *isoAsp* not seen in *Asp* [61]. These reporter ions are a result of an alternate cleavage due to presence of the extra methylene group in the peptide backbone of *isoAsp* (Figure 7). The *c*_n−1_ + 57 and z_m-n + 1_^●^ −57 are a unique complementary pair formed as a result of the C_α_ and C_β_ bond cleavage. 

Representative TICs of the tryptic peptides are provided (Figure 8A) to demonstrate the confirmation of a primary sequence as well as to assess PTMs and their specific loci. A confirmatory sequence for each conformer was determined with 100% coverage, i.e., complete *c* and *z* ion detection (Figure 8B). For purpose of clarity, the *c* ion series are shown for the *isoAsp* containing fragment (Figure 8B top panel—red spectrum), and the *z* ion series are shown for the *Asp* containing fragment (Figure 8B bottom panel—blue spectrum). To clarify the *Asp* assignments, the spectra were probed for the reporter ion previously discussed. The *z*_4_ –57 was present in the tryptic peptide from PcaTX-1a and not detected in PcaTX-1b (Figure 8C). Similarly, the presence of the reporter ion *c*_8_ + 57 seen in the tryptic peptide from PcaTX-1a is not present in the later eluting tryptic peptide from PcaTX-1b (Figure 8D). There were other tryptic peptides with *Asp*, but only the *Asp* at the C-terminus region displayed the diagnostic reporter ions as described (data not shown). Taken together, these data suggest the differences in the overall primary structure of PcaTX-1a and PcaTX-1b were due to the alternate isomers of *Asp*^33^.

Previously, we described PcaTX-1 conformers in terms of ATD in TWIMS and did not detect an observable difference in the overall “shape” of PcaTX-1a and PcaTX-1b. To probe differences in the tryptic peptides for each conformer, EIMs of the C-terminus are provided (Figure 9A). Like the multiply charged state ions of the parent peptide, the multiply charged states of the tryptic peptides for PcaTX-1a and PcaTX-1b did not differ significantly: 379 and 381 Å^2^, respectively. However, the *isoAsp* and *Asp* residues from that peptide did alter the mobility of the singly charged ions that yielded different CCSs and provided further clarification of the isomeric pair. The CCS of the tryptic peptide from PcaTX-1a of 347 Å^2^ was significantly different from CCS of the PcaTX-1b of 356 Å^2^. This data suggests that the *isoAsp* causes a regional change in the peptide structure due to tighter folding in the C-terminus loop that doesn’t affect the overall observed shape of the peptide.

### 2.5. Using ETD vs. ETcaD

Although ETD can deliver unique fragmentation patterns for de novo sequencing, there is a challenge in this application to normal proteomic workflows. As the precursor *m/z* increases, there is a decrease in fragment ion yield due to the charge-reduced electron transfer of the intact peptide; a phenomena pronounced with doubly charged peptide precursor ions above 400 *m/z* [62]. Trypsin is one of the most common enzymes used in bottom-up proteomics and results in an average digestion product of 700–1500 Da [63,64]. Therefore, due to the size of these products, singly and doubly charged species dominate the overall landscape of the peptide profile [65]. One solution to address this issue is the use of supplemental activation (ETcaD) in the transfer that reportedly augments the *c* and *z* ion abundance. Although our studies provided an adequate signal for [M + 3H]^3+^ charged states and greater, we collected data from the tryptic digests previously described (Figure 8) and altered the MS protocol to apply a small voltage (~7–15 V) to the transfer lens that channels ions to the ToF flight tube. Product ion spectra of three acquisitions have been provided for comparison: ETD on the [M + 2H]^2+^, ETcaD on the [M + 2H]^2+^, and ETD on the [M + 3H]^3+^ (Figure 10A). All three spectra have been magnified 75X from the original base peak to demonstrate the poor fragmentation efficiency for both ETD and ETcaD on [M + 2H]^2+^. Enlargement of the spectra has been provided in both the low mass range of 100–325 *m/z* (Figure 10B) and a mid-mass range of 750–950 *m/z* (Figure 10C). Since high mass range fragments were seen in all three fragmentation techniques and are the easiest to form, we selected the low and mid mass ranges to analyze the effect and putative benefit of using supplemental activation. In all mass ranges observed, the fragmentation efficiency of ETD on the [M + 2H]^2+^ was very low but did provide classical *c* and *z* ions for confirmatory de novo sequencing. The reporter ions for *isoAsp* were not observed in our experiments (data not shown). However, the ETcaD spectra of the [M + 2H]^2+^ displayed a very convoluted “hybrid” spectra of *b*, *c*, *y*, and *z* ions. Supplemental activation did not enhance the sequencing process, nor did it provide additional *c* and *z* ions. As in the ETD spectra, the reporter ions for *isoAsp* were not observed in the ETcaD experiments (data not shown). We tried this approach on ten other peptides of different residue makeup from these experiments with a mix of results (data not shown). However, many of the analyses yielded a similar assessment as to the one provided here. One last remark is the addition of <5 V of supplemental activation on classical ETD experiments on the [M + 3H]^3+^ ions (or ions of greater charge) did enhance the fragmentation up to 15% increase, but did not alter the quality of the fragmentation products, i.e., it did not form *c* and *z* ions that were not detected prior to the additional collision energy addition. Taken together, these data suggest that the use of supplemental activation is peptide specific and should be used with caution due to the formation of hybrid spectra that may convolute the experimental desire.

### 2.6. Multiple Enzymes Digestion Products

Even with proper residue assignments as described above, 100% coverage of a peptide is usually very difficult as it is dependent on the overall nature of the primary structure. Some of the enzyme cleavage products are at a low abundance or not chromatographically favorable. Therefore, to expand the coverage of all possible sequence scenarios, multiple enzymes were employed with different cleavage patterns [66]. Both PcaTX-1a and PcaTX-1b were sequenced with an overlay scheme to confirm the final primary structure assignment (Figure 11).

Trypsin cleaves on the carboxyl side of the amino acids *Lys* and *Arg*. Chymotrypsin preferentially cleaves on the carboxyl side of hydrophobic amino acids *Tyr*, *Trp*, and *Phe* and at a slower rate, it can hydrolyze amide bonds of *Leu* and *Met*. Discrimination between *Leu* and *Ile* was achieved using AccQ●Tag chemistry on the enzymatic peptides.

Asp-N endoproteinase hydrolyzes peptide bonds on the N-terminal side of aspartyl residues, but cannot cleave a beta peptide linkage of *isoAsp* due to an extra methylene group in the backbone [67]. Since both analogs displayed the same primary structure, but had different Asp-N digest products, we confirmed our hypothesis from MS analyses that the later eluting analog of PcaTX-1b had three *Asp* residues in its primary structure while PcaTX-1a possessed an *isoAsp* in position 33. 

One final note for consideration was the proper assignment of disulfide bridges. This assignment is an important PTM and needed to establish and maintain physiologically optimized tertiary structures [68]. After the sequence of PcaTX-1a and PcaTX-1b had been determined, we assumed that the *Cys* bridges were conserved and followed the same pattern of linkages as seen in most related theraphosids, i.e., C3:C17, C10:C22, and C16:C30. To confirm this hypothesis, we used two analytical approaches by varying reduction, enzyme specificity, and fragmentation formats (data not shown) [69,70]. 

## 3. Discussion

Over the last fifty years, venoms have been a rich source for bioactive molecules [71]. The early studies of toxins from snakes, scorpions, and spiders and their ability to target the circulatory and nervous systems have expanded to more neglected species and diverse targets [72]. These studies have exploded into a unique and interdisciplinary field, referred to as toxinology, dedicated to the biochemistry, pharmacology and toxicology of toxins and poisons, from plants, bacteria, and animals. With over 220,000 species of venomous animals, there is a seemingly unlimited number of possible candidates for investigation [73]. As an example, spider venoms are particularly attractive with an estimated 18 million peptides from their venoms [74]. With this voluminous accumulation of data, and an even larger repository of yet uncharacterized molecules, it was a reasonable discourse for researchers to invest in their therapeutic potential. However, the elucidation of these activities has proven to be very complex, and in many cases, enigmatic [75]. With the multitude of potential candidates, less than ten have yielded venom-derived drugs [73]. As a result, there has been a steady decline in natural product research in the pharmaceutical industry. The perception is “the easy compounds have been made, the progress of research is too slow (even in high throughput screening), and the remainder are too difficult.” [76]

Our laboratory solution is “de novo”. From the Latin meaning “of new,” it is more directly translated to mean “from the beginning.” Current data suggests that these peptides have evolved from a conserved structural framework that can explain their pharmacokinetic and pharmacodynamic properties [77]. There are many challenges in elucidating this framework, including, but not limited to, the promiscuity of toxin activity, lack of industry standardization for physiological assays, and the overall diversity of their primary structures. Therefore, we need to start from the beginning and follow the architectural principle: “form follows function.” Pharmacologists generally agree that the key to unraveling the potential of venom peptides is a closer examination of the structure-activity relationship with a “particular emphasis on how changes in the molecular structure of drugs and receptors/channels result in kinetic changes in the function of receptors/channels” [78]. This, coupled to the fact that neurotoxins elicit their primary effects on either ion permeation or voltage-dependent gating [79], makes the study of PTMs and their respective analogs a powerful first step. To further unravel structural diversifications, HDMS^E^ with ETD provides a standardized primary analytical technique.

A challenge in primary structure elucidation is the biochemical instability inherent in proteins. Deamidation of *Asn* and the isomerization of *Asp* is one of the most common non-enzymatic PTMs [80,81,82]. *isoAsp* has been implicated in several high-profile pathologies, e.g., Alzheimer’s disease and metastatic breast cancer, due to its aggregation or loss of functionality [67]. We employed HDMS^E^ to screen venoms and to discriminate between *Asp* and *isoAsp*. We characterized the two most abundant isoforms of PcaTX-1, a 36-residue peptide with the primary structure of NH_2_-AGCKYLFGSCKEDSDCCKHLGCRRKAPQYCGW*D*GTF-NH_2_, found in five of ten tarantula species from the genus *Poecilotheria*. The difference in these isobaric conformers is the *Asp*^33^ residue: *isoAsp*^33^ in PcaTX-1a and *Asp*^33^ in PcaTX-1b. 

The isomerization of *Asp*^33^ in PcaTX-1b yields a modified structure due to the beta amino acid *isoAsp*^33^ in PcaTX-1a. Rearrangement occurs most commonly in -*Asp*-*Gly*- and -*Asp*-*Ser*- sequences when these residues are in a flexible region of the peptide due to a lack of side-chain interference. Although this subtle modification is difficult to identify, the substitution has profound effects on the molecule’s biological structure and concomitant activity. Specifically, the extra methylene group in the peptide backbone allows for more degrees of rotational freedom than an alpha peptide linkage; the amide plane has a hybrid resonance that restricts movement to the C-N bond (*Ф* angle) and C-C bond (*ψ* angle). The direct consequence of the rigid plane found in normal alpha residue linkage is the formation of predictable secondary structures: α-helix and β-sheet. With the gain of this rotational freedom in the C-terminus, the properties of PcaTX-1a are significantly altered. The partition coefficient (*k*’) of PcaTX-1a decreased by 9.3% as compared to PcaTX-1b (Figure 3A) suggesting the overall solubility has increased. The digestion peptides that contained these residues showed a similar loss of hydrophobicity (Figure 6A). 

A more intriguing aspect of this isomerization is the effect it has on the tertiary structure of the peptide and consequently its ion channel modulations. The beta amino acid lengthens the backbone of the C-terminal portion of the peptide. Although the resolution of our IMS experiments did not show a significant difference in the peptide conformer’s mobility (Figure 4), a change in the tryptic peptide’s mobility was used as a confirmation in the de novo process to clarify the *Asp*^33^ isomerization (Figure 9). Taken together, these data suggest a subtle change on the C-terminus may account for the solubility differential as well as the modulatory effects seen in the electrophysiological experiments (Figure 5). Our laboratory has examined other isobaric conformers with *Asp* isomerization and found a diminished activity in most of the peptides studied so far (data to be published). One item of note is we were unable to find an analog of PcaTX-1 with *Asn*^33^, suggesting that the *Asp*’s seen are not the result of deamidation. While we were unable to locate an *Asn*33*Asp* in any of the venoms, we do not rule out the possibility of deamidation occurring during the in vivo production of these toxin constructs. We were also unable to produce an *Asn* analog in the peptide under neutral pH, or pH of the digestion conditions. 

An explanation for the presence of the *isoAsp* is the repair system facilitated by PIMT. In vitro aging of proteins under physiological pH and temperature show a marked increase in *isoAsp* formation [83]. However, in vivo experiments have demonstrated the PIMT mediated catalysis is essential for reduction in the accumulation of the *isoAsp* residues [84]. Since the energy requirement for repair is considerably less than the degradation and synthesis of a new protein, it is essential that this enzyme functions properly [85]. If the venom is stored for a prolonged period, the isomerization could be occurring without repair. An alternate explanation is that the specimen is aging, and the catalysis process is less efficient. A more intriguing possibility is the PIMT catalysis is used to make *Asn-*to-*Asp* substitutions exchanging a polar uncharged residue (amidic) to a polar negatively charge (acidic) residue. While both deamidation of *Asn* and isomerization of *Asp* lead to the formation of *isoAsp,* the action of PIMT will restore the normal peptide backbone but not the *Asn* side chain [84]. Therefore, three possible outcomes of these two modifications are: (1) the restoration of the *Asp* and restored “normal” function, (2) an *Asn-*to-*Asp* substitution that has an altered but useful function, or (3) an efficient protease degradation since there is no known protease that recognizes the *isoAsp*-X peptide bond.

Like many α-scorpion toxins that bind to site 3 and β-scorpion toxins that bind to site 4, both PcaTX-1a and PcaTX-1b conformers modulate the inactivation kinetics of VGSCs by increasing the time constants of fast inactivation with the latter peptide shifting the voltage dependence of activation in a depolarizing, not hyperpolarizing direction. While our data suggest two distinct binding sites with differential affinities, it is unclear whether the modulation seen by either conformer is due to interactions at site 3 and/or site 4, respectively. An alternate possibility is they bind to an unidentified region of DIV S4 similar to δ-conotoxins, i.e., site 6 [28]. The positively charged residues of S4 are interspaced with two hydrophobic residues which are drawn by negatively charged residues in adjacent segments to stabilize S4 against the pull by the negative membrane potential [86]. Depolarization causes the S4 segment to move outward initiating the open conformation, but inactivation may be coupled to activation, working in contrast to the “classical” Hodgkin-Huxley postulate [86]. More extensive testing is needed to elucidate the binding and subsequent modulations caused by the presence of the *isoAsp* and may provide a contrasting model for clarification.

## 4. Conclusions

In this manuscript, we have provided an example of an analytical process to demonstrate a standardized approach to primary structure and PTM illumination. We described two novel VGSC modifiers, PcaTX-1a and PcaTX-1b, using traditional UPLC-HDMS combined with CID and ETD fragmentation techniques. This approach allows for complete clarification in our de novo process. With the added dimension of IMS, a clearer picture of SAR for toxin activity is possible.

## 5. Materials and Methods 

### 5.1. Reagents and Chemicals

All reagents utilized were purchased from Fisher Scientific (Waltham, MA, USA) or Sigma-Aldrich (Saint Louis, MO, USA) as ACS reagent grade or better; LC/MS grade acetonitrile was used for high-resolution mass spectrometric determinations.

### 5.2. Venom Collection and Preparation

Specimens of *Poecilotheria* were housed in a vivarium at Carbon Dynamics Institute, LLC (Sherman, IL, USA) at a temperature 23–27 °C, humidity 65%–75%, and fed a variety of live crickets, roaches, or locusts. A specimen was anesthetized with CO_2_ until areflexia was observed. A Grass SD9 Square Pulse Stimulator was used to apply a 2–5 V potential field (2 Hz; 2 ms) to the dorsal base of each chelicera. Venom was collected directly into an Eppendorf LoBind^®^ tube and freeze dried at −50 °C at a vacuum of 0.05 mBar in a Labconco FreeZone Freeze Dry System (Labconco, Kansas City, MO, USA) to dryness. The venom was stored at −25 °C until needed for subsequent analyses.

### 5.3. Size Exclusion Chromatography (SEC) Purifications

To purify a peptide toxin, ~5 mg of freeze-dried venom was suspended into the initial mobile phase buffer and injected into a Bio-Rad BioLogic DuoFlow High Performance Liquid Chromatograph (HPLC) (Bio-Rad, Hercules, CA, USA) equipped with a Phenomenex Yarra SEC-2000 3 μm 7.8 × 300 mm column at a flow rate of 1.0 mL/min (mobile phase 0.1M NH_4_HCO_3_:20% acetonitrile). The effluent stream from the chromatographic separation was coupled to a Bio-Rad BioLogic QuadTec UV-Vis Detector (Bio-Rad, Hercules, CA, USA) and a Bio-Rad BioLogic Fraction Collector (Bio-Rad, Hercules, CA, USA). The fraction collection was adjusted to collect 0.5 mL fractions. Fractions were freeze-dried at −50 °C at a vacuum of 0.05 mBar in a Labconco FreeZone Freeze Dry System and stored at −25 °C until needed for subsequent analyses (see Appendix A).

### 5.4. Reversed-Phase Chromatography (RPC) Purifications

To further purify a toxin for biochemical and physiological determinations, SEC fractions were suspended into the initial mobile phase buffer and injected into a BioRad BioLogic DuoFlow High Performance Liquid Chromatograph (HPLC) equipped with a Phenomenex Aeris PEPTIDE SB-C18 5 μm 10 × 250 mm column at a flow rate of 5.0 mL/min, with a 2.00 min hold at 90:10 A:B (mobile phase A = 0.01% formic acid; mobile phase B = acetonitrile) followed by a binary mobile phase gradient to 10:90 A:B in 30.00 min at a flow rate of 4.0 mL/min. The effluent stream from the chromatographic separation was coupled to a BioRad BioLogic QuadTec UV-Vis Detector and a BioRad BioLogic Fraction Collector. The fraction collection was adjusted to collect 0.5 mL fractions. Fractions were freeze dried at −50 °C at a vacuum of 0.05 mBar in a Labconco FreeZone Freeze Dry System and stored at −25 °C until needed for subsequent analyses (see Appendix A).

### 5.5. Reversed-Phase Chromatography (RPC) Determinations

To further characterize a toxin or a toxin enzyme digest for biochemical determinations, RPC fractions or digests were suspended into the initial mobile phase buffer and injected into a Waters Acquity^®^ Ultra Performance Liquid Chromatograph (UPLC) (Waters Corporation, Milford, MA, USA) equipped with an Acquity UPLC BEH C18 1.7 μm 2.15 × 50 mm column at a flow rate of 0.50 mL/min, a 1.00 min hold at 90:10 A:B (mobile phase A = 0.10% formic acid; mobile phase B = acetonitrile) followed by a binary mobile phase gradient to 60:40 A:B in 40.00 min, then 15:85 A:B in 2.00 min, lastly held for 7.00 min providing proper peak shape, separation, and reduction of interferences (see Appendix A).

### 5.6. Toxin Digestions and Modifications

To determine the overall amino acid composition of a peptide, acid hydrolyzed peptides were diluted with 0.1 M HCl prior to derivatization. In general, 60 μL of AccQ●Fluor Borate Buffer were added to each diluted hydrolysate followed by 20 μL of the AccQ●Fluor Reagent. The samples were heated for 10 min at 55 °C, and 5 μL aliquots were injected onto a Waters AccQ●Tag column (3.9 × 150 mm) for optimal separation. Derivatized amino acids were detected by a 470-scanning fluorescence detector [87,88]. 

All protein digestions were performed with slight variations to protocols provided by Thermo Scientific (Thermo Fisher Scientific, Waltham, MA, USA) [89].

### 5.7. Accurate Mass Determinations

Any aliquot of venom, toxin, or peptide digests were injected into a Waters UPLC Acquity^®^ (Waters Corporation, Milford, MA, USA) interfaced with a Synapt^®^ G2-Si HDMS (Waters Corporation, Milford, MA, USA) equipped with an ESCI^®^ z-spray atmospheric pressure ionization source. All acquisitions were made with lock mass correction using [Glu^1^]-fibrinopeptide B human as a reference. To maximize ion resolution, all samples were processed for de novo sequencing and molecular mass characterization in “W-ion” optics mode (W-Optics^TM^) (Waters Corporation, Milford, MA, USA) that provided FWHM >50,000 resolution. The following MS conditions were routinely employed: polarity ES^+^, capillary 2.80 kV, sampling cone 30, source offset 30, ToF 10.00 kV, pusher cycle time 124 μs and pusher frequency 8065 Hz, cone gas 600 L/h, source 125 °C, and desolvation 350 °C.

### 5.8. Collision Induced Dissociation (CID) and Electron Transfer Dissociation (ETD)

To ensure efficient acquisition of all peptides and/or co-eluting compounds, all analyses for de novo sequencing were performed in data directed analysis (DDA^TM^) focused upon only those precursor ions that met strict selection criteria, i.e., signal intensity, charge state, and exact mass, allowing them to be selected preferentially for MS/MS. Routinely, four consecutive 0.3 s MS-MS scans were acquired: (1) CID on [M + 1H]^1+^ (CE Ramp 20–30 V), (2) CID on [M + 2H]^2+^ (CE Ramp 20–45 V), (3) electron transfer dissociation with supplemental activation (ETcaD) on [M + 2H]^2+^ (wave velocity 600 m/s, wave height ~0.23 V, transfer CE 5–15 V), and (4) ETD on [M + 3H]^3+^ (wave velocity 600 m/s and wave height ~0.23 V). All ETD experiments were acquired with a discharge current of 24 mA after tuning the reagent ion 128 *m/z* (1,3-dicyanobenzene) in the negative chemical ionization (nCI) source to ~1e^5^ intensity. Refill parameters were set as follows: refill time (0.1 s refill interval and 1.0 s refill interval).

### 5.9. Traveling Wave Ion Mobility Spectrometry (TWIMS)

TWIMS was performed as described [90,91]. MS conditions used: capillary voltage 1.0 kV, IMS wave velocity 650 m/s (ramp 200–850 for optimum separation were used), wave height 40 V, trap/transfer gas pressure ~2.0e^−2^ mbar (Ar), IMS gas pressure 1.0 mbar (N_2_), and a 7 V pulse adjusted as needed in subsequent experiments to maximize arrival time distribution (ATD). 

### 5.10. Cell Culture

Catecholamine A-differentiated (CAD) cells were maintained as previously described [92].

### 5.11. Whole-Cell Recording

Whole-cell voltage-clamp recordings were performed as previously described [92] with the following changes: HEKA PatchMaster V 2x90.5 (HEKA Instruments Inc., Holliston, MA, USA) and *V*_H_ of −100 mV.

### 5.12. Data Analysis and Statistics

Mass spectra and electrophysiological traces were processed as previously described [92] and all MS spectra are shown as monoisotopic masses with the following additions.

Estimated CCS measurements were made by subtracting the instrumental offset and calculating the correct drift time (*t*_d’_):(1)$$t_{d\prime }=t_{d}-\left(\frac{c\sqrt{\frac{m}{z}ion}}{1000}\right)ms$$
where *t*_d_ is the drift time of ion and *c* is a correction factor set-up for each instrument. Published CCS data (Ω) was then corrected by using a normalized CCS (Ω’) that accounted for reduced mass and charge state:(2)$$\Omega \prime =\left(\frac{\Omega x\sqrt{\mu }}{z}\right)$$

A plot was generated for *t*_d’_ versus Ω’ that converted experimental drift times to estimated CCS [93]. 

The time constant (τ_f_) for fast inactivation was determined from an exponential curve and computed in two steps using the HEKA FitMaster V 2x90.5: (1) semi-logarithmic regression of *x* vs ln(*y*) and (2) abs(1/slope).

### 5.13. Accession Number

Database: UniProt Knowledgebase under the accession number(s) C0HLN4.

## Figures and Tables

**Figure 1 toxins-12-00207-f001:**
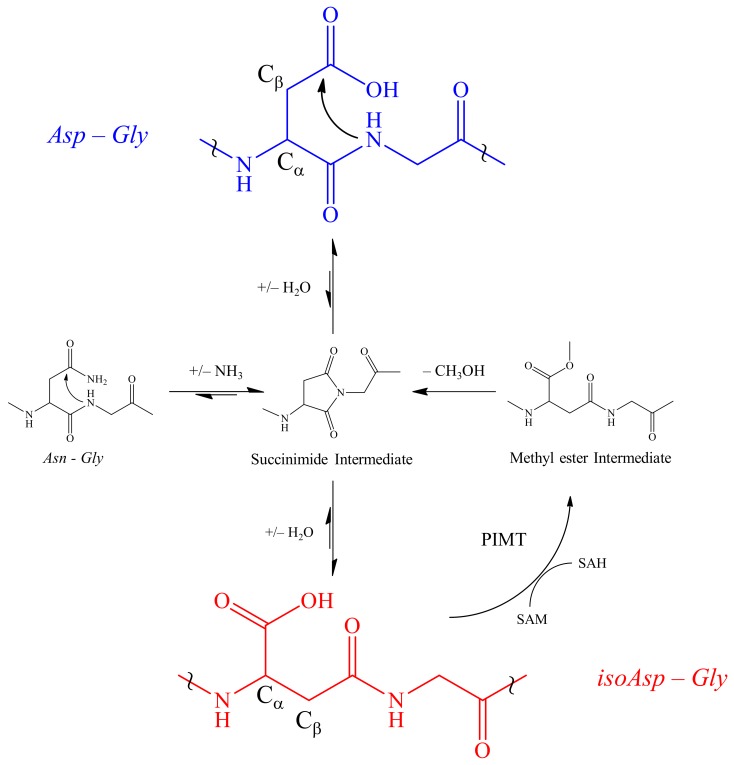
Spontaneous deamidation of *Asn* and isomerization of *Asp*. Deamidation of *Asn* and/or dehydration of *Asp* (blue) leads to the formation of a five membered L-succinimide ring intermediate due to a nucleophilic attack by the glycyl amine on the carbonyl of the *Asp* R group. Subsequent hydrolysis may revert to the *Asp* or more commonly leads to the formation of an *isoAsp* (red). Following isomerization, protein-isoaspartyl methyltransferase (PIMT) repairs the damaged *Asp* by transferring a methyl group from S-adenosylmethionine (SAM) to the carboxylic acid of *isoAsp* forming the methyl ester intermediate which can be hydrolyzed leaving a S-adenosylhomocysteine (SAH). The intermediate is then hydrolyzed back into the succinimide intermediate. At a significantly lower level, racemization of the L-succinimide leads to small levels of D-succinimide and subsequent D enantiomers of *Asp* and *isoAsp* (not shown). The symbol ~ is used to focus on a region of the peptide backbone.

**Figure 2 toxins-12-00207-f002:**
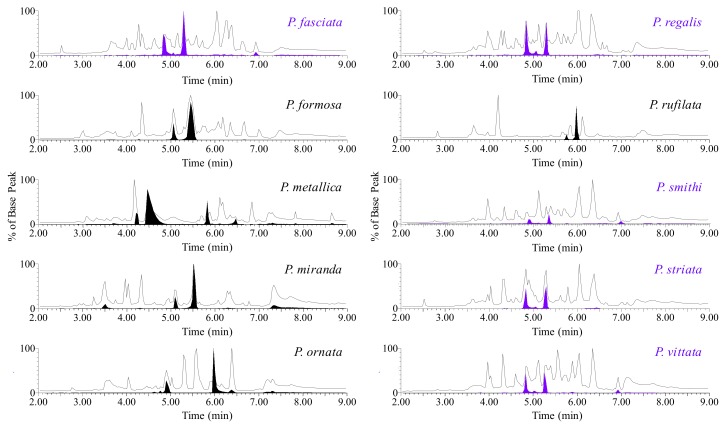
Representative TICs of *Poecilotheria* venom. A reversed-phase liquid chromatographic separation of venom from ten species of *Poecilotheria* reveals a complex and diverse mixture of molecules primarily made up of small peptides ranging from 3–6 kDa. A filled EIC is overlaid for each species demonstrating the presence of chromatographically separable isobaric conformers. The venom of five species displayed a conserved conformer group (EIC filled in purple) with an average molecular mass of ~4027.6 Da.

**Figure 3 toxins-12-00207-f003:**
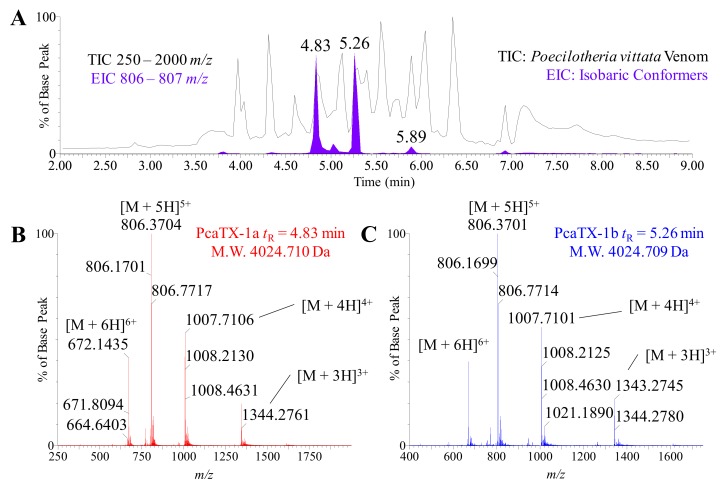
Representative high-resolution mass spectra of PcaTX-1. (**A**) The TIC for *P. vittata* venom with a range from 250–2000 *m/z* is shown in the background for reference to reveal the EIC range 803–807 *m/z* isobaric conformers (filled purple). There were a total of seven peaks seen in the EIC, however, only three were isobaric conformers based on mass determinations (see text for detail). (**B**) A multiply charged electrospray spectrum of PcaTX-1a at *t*_R_ 4.83 min and (**C**) the multiply charged electrospray spectrum of PcaTX-1b at *t*_R_ 5.26 min revealed a typical charge envelope of a compact peptide. The [M + 5H]^5+^ ion and the [M + 4H]^4+^ clusters are identified. The spectrum (FWHM >50,000) were processed and determined to have molecular monoisotopic masses of 4024.710 and 4024.709 Da, respectively. The purple color used indicates the presence of either an *isoAsp* or an *Asp* that have not been differentiated. The red color used indicates the presence of *isoAsp* and the blue color used indicates the presence of *Asp*.

**Figure 4 toxins-12-00207-f004:**
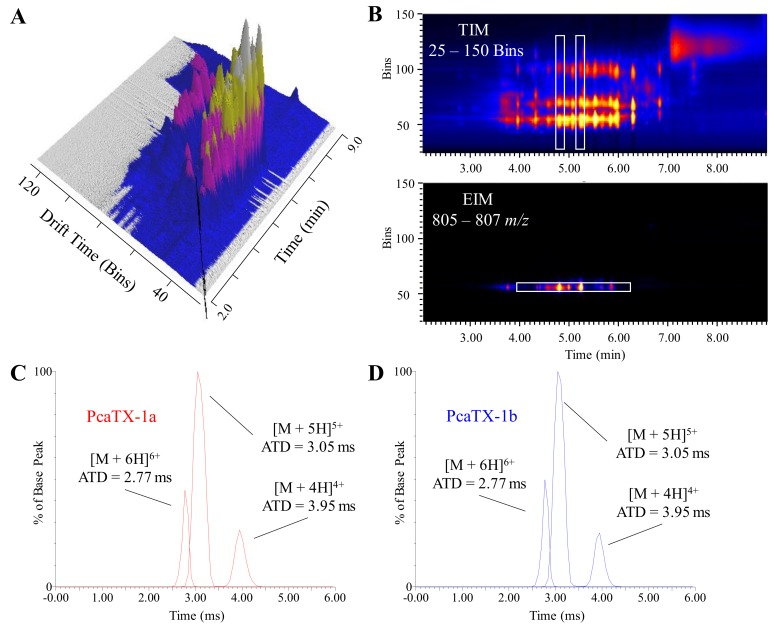
Representative ion mobilograms from *P. vittata* venom. (**A**) A three-dimensional image showing a two-dimensional chromatogram of *P. vittata* venom coupled with its corresponding total ion mobilogram (TIM) demonstrates the additional level of concomitant data that may be probed for investigation. (**B**) The presence of ions with different mobilities, i.e., based on charge, size, and shape, at a specific retention time can be seen in a two-dimensional heat map of retention vs. mobility (top panel). A white rectangle has been placed at *t*_R_ 4.83 min and 5.26 min corresponding to the isobaric conformers described in Figure 3. All of the masses were calculated to determine if the ions were co-eluting molecules and/or different charge states of the same molecule. To further examine the mobility of PcaTX-1a as it relates to PcaTX-1b, a white rectangle has been placed around the extracted ion mobiligram (EIM 805–807 *m/z*) of the [M + 5H]^5+^ (bottom panel) which revealed no significant difference in arrival time ditsruibutions (ATDs). The EIM of the 6^+^, 5^+^, and 4^+^ charge states for PcaTX-1a (**C**) and PcaTX-1b (**D**) revealed no significant difference in ATDs for the two conformers at *t*_R_ 4.83 min and 5.26 min (Figure 3A). The red color used indicates the presence of *isoAsp* and the blue color used indicates the presence of *Asp*.

**Figure 5 toxins-12-00207-f005:**
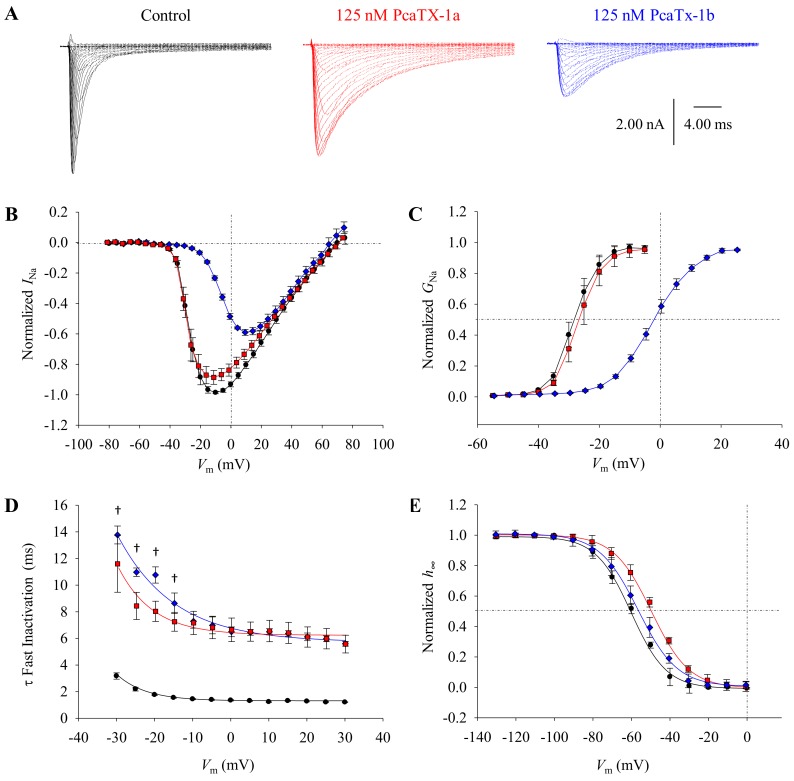
The effect of PcaTX-1 conformers on Na_v_1.7 currents. (**A**) Na_v_1.7 currents were elicited by a 5 mV (50 ms) depolarization step from −75 to +75 mV from a *V*_H_ of −100 mV before (black traces) and after perfusion of 125 nM PcaTX-1a (red traces) and PcaTX-1b (blue traces). (**B**) Current-voltage (*I/V*) curves (normalized to the control) before (●) and after treatment of PcaTX-1a (red ■) and PcaTX-1b (blue ♦) are shown and were fitted to evaluate the *V*_½_ and *V*_rev_. While PcaTX-1b showed a significant shift in *V*_½_ as compared to both the control and PcaTX-1a, neither peptide caused a significant shift in the *V*_rev_ compared to the control. (**C**) Normalized to maximum conductance, conductance-voltage (*G/V*) curves for Na_v_1.7 currents before (●) and after treatment of PcaTX-1a (red ■) and PcaTX-1b (blue ♦) are shown and fitted to a Boltzmann distribution. The *G V*_½_ for the control did not significantly shift in the presence of PcaTX-1a but displayed a significant depolarizing shift of ~22.2 mV in the presence of PcaTX-1b compared to the control. There was no significant difference in the *G k* for the control and the PcaTX-1a treatment, however, PcaTX-1b significantly shifted the slope as compared to both the control and PcaTX-1a. (**D**) Time constants (τ) of fast inactivation were plotted against voltage (τ/*V*) before (●) and after treatment of PcaTX-1a (red ■) and PcaTX-1b (blue ♦). Both PcaTX-1a and PcaTX-1b treatments caused a significant shift in the τ at all voltages as compared to the control, while the dagger (†) denotes where PcaTX-1a is significantly different than PcaTX-1b. (E) Steady-state fast inactivation (*h_∞_*) was determined using a standard two-pulse protocol. At a *V*_H_ of −100 mV, conditioning pre-pulses of 50 ms durations ranging from −130 to +5 mV in 10 mV steps were delivered, and currents were measured at a depolarization step to 0 mV pulses (20 ms). Normalized to the control, peak Na_v_1.7 currents obtained from the 20 ms test pulse to 0 mV following the pre-pulse are plotted against the pre-pulse potentials. Curves were fitted to a Boltzmann distribution. The *h_∞_ V*_½_ before (●) and after treatment of PcaTX-1a (red ■) showed a significant difference, while there was no significant difference between the control and PcaTX-1b (blue ♦). However, there was a significant difference in *h_∞_ V*_½_ between PcaTX-1a and PcaTX-1b. There was no significant difference in the *h_∞_ k*.

**Figure 6 toxins-12-00207-f006:**
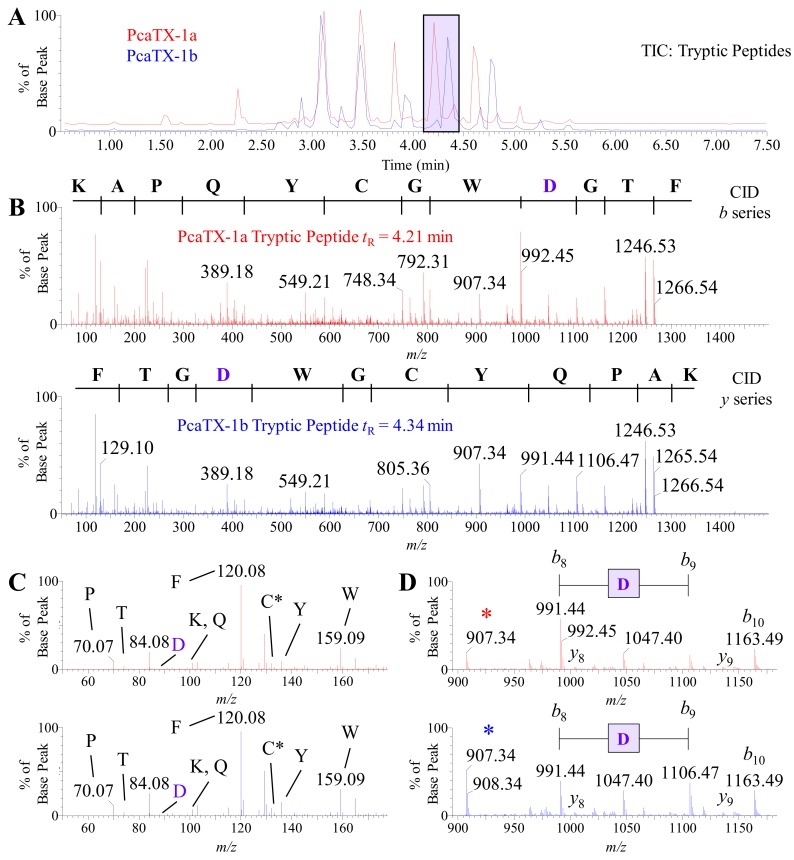
Representative TIC and CID spectra of PcaTX-1 tryptic peptides. (**A**) Representative TICs for the tryptic peptides PcaTX-1a (red) and PcaTX-1b (blue) are overlaid to show the differences in the trypsin fragment profiles. These differences in peptide elution, as seen in the purple rectangle centered at ~*t*_R_ 4.28 min, suggested differences in primary structure. (**B**) The doubly charged parent ion (mp^●+^) of 714.80 *m/z* for both PcaTX-1a tryptic peptide (*t*_R_ = 4.21 min) and PcaTX-1b tryptic peptide (*t*_R_ = 4.34 min) were passed through a charged transfer chamber filled with argon gas that induced low energy collisions (CE 29) resulting in subsequent fragmentation along the N-C bond in the backbone to provide MS/MS spectra with unique and characteristic *b* and *y* daughter ions (md^●+^) used for preliminary de novo sequencing. For clarity, a *b* series is shown for the PcaTX-1a tryptic peptide and a *y* series for the PcaTX-1b tryptic peptide. (**C**) A unique feature of Q-ToF MS in low energy CID is the formation of internal immonium ions. The low range mass spectra (50–180 *m/z*) for both tryptic peptides confirms the presence of nine residues for the putative sequences determined from the daughter ion spectra; the only immonium ions not detected were for *Gly* and *Ala* since their masses are below the mass range cutoff for these experiments. The immonium for *Cys* is the carbamidomethylated ion at 133.04 *m/z*. (**D**) Biemann *b* and *y* ion assignments were made for all residues but identified here for a small range of the spectra for clarity (89–1185 *m*/*z*). The shift from 991.44 *m/z* to 1106.47 *m/z* showed a mass defect of 115.03 Da suggesting an *Asp* residue (purple), but cannot distinguish between isomeric configurations. Subtle differences in ion abundance of the *b*_8_ ion as well as the internal acyl fragment at 907.34 *m/z* (*) suggest differences in cleavage at this location (see text for details). The red color used indicates the presence of *isoAsp* and the blue color used indicates the presence of *Asp*.

**Figure 7 toxins-12-00207-f007:**
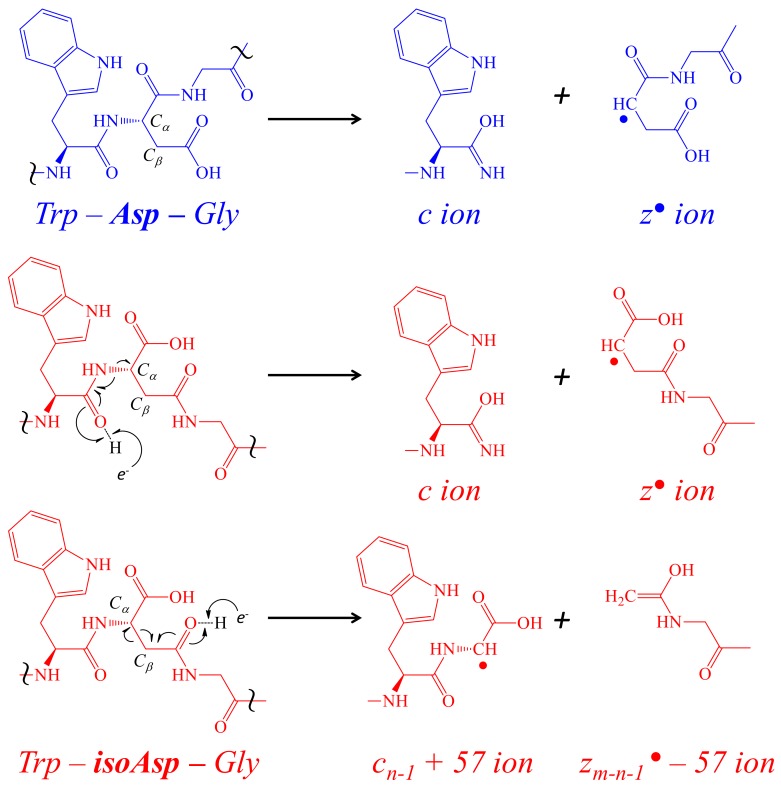
The formation of *c* and *z* ions with corresponding reporter ions from *isoAsp*. The formation of *c* and *z* ions in ETD is due to the transfer of an *e*^–^ donated by a radical negative anion (1,3-dicyanobenzene or nitrosobenzene) resulting in the cleavage of the peptide backbone on the amino side of a residue; the cleavage between the *Trp* and *Asp* are shown here. When an extra methylene group is present in the peptide backbone due to the isomerization of *Asp* into *isoAsp*, an alternate cleavage is possible forming reporter ions *c_n−1_* + 57 and *z_m-n−1_*^●^ −57 ions, where the *m* is the total number of residues and the *n* is nth position of *Asp*. The red color used indicates the presence of *isoAsp* and the blue color used indicates the presence of *Asp*. The symbol ~ is used to focus on a region of the peptide backbone.

**Figure 8 toxins-12-00207-f008:**
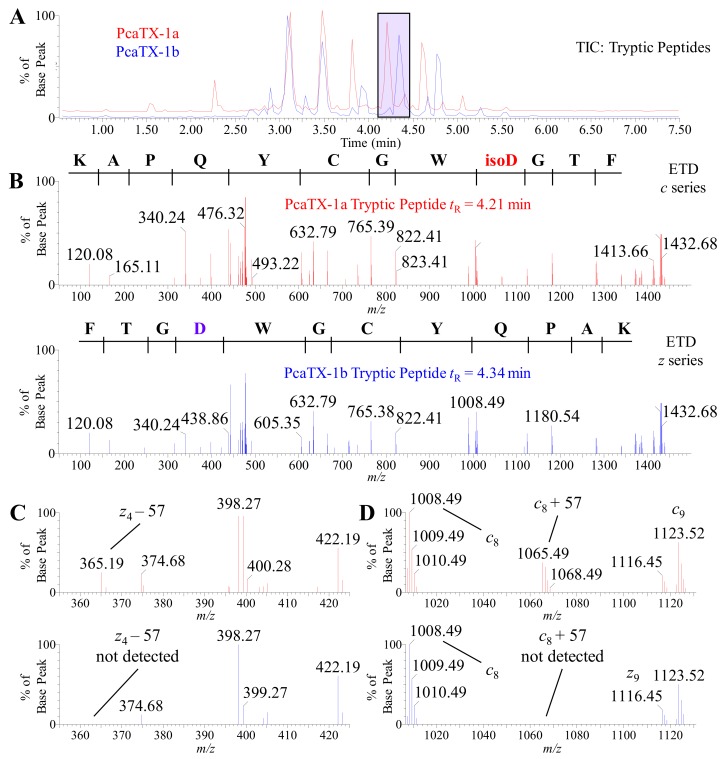
Representative TIC and ETD spectra of PcaTX-1 tryptic peptides. (**A**) Representative TICs for the tryptic peptides PcaTX-1a (red) and PcaTX-1b (blue) are overlaid to show the differences in the trypsin fragment profiles. These differences in peptide elution, as seen in the purple rectangle centered at ~*t*_R_ 4.28 min, suggested differences in primary sructure. (**B**) The triply charged parent ion (mp^●+^) of 476.88 *m/z* for both PcaTX-1a tryptic peptide (Figure 8A: *t*_R_= 4.21 min) and PcaTX-1b tryptic peptide (Figure 8A: *t*_R_ = 4.34 min) were isolated for a gas-phase ion/ion interaction with radical anions (1,3-dicyanobenzene) for electron transfer and subsequent fragmentation along the N-C_α_ bond in the backbone to provide MS/MS spectra with unique and characteristic *c* and *z* daughter ions (md^●+^) used for confirmatory and complete de novo sequencing. For clarity, a *c* series is shown for the PcaTX-1a tryptic peptide and a *z* series for the PcaTX-1b tryptic peptide. (**C**) and (**D**) One aspect unique to ETD used for distinguishing *Asp* and *isoAsp* is the presence of the reporter ions *c_n−1_* + 57 and *z_m-n−1_*^●^ −57 (see Figure 7). The presence of reporter ions *c*_9_ + 57 and *z*_4_^●^ −57 were detected in the trypsin fragment PcaTX-1a (red), but not in PcaTX-1b (blue). These data confirmed the presence of *isoAsp*^33^ in PcaTX-1a and *Asp*^33^ in PcaTX-1b and suggested differences previously described were due to this subtle structural modification of the two isobaric conformers. The red color used indicates the presence of *isoAsp* and the blue color used indicates the presence of *Asp*.

**Figure 9 toxins-12-00207-f009:**
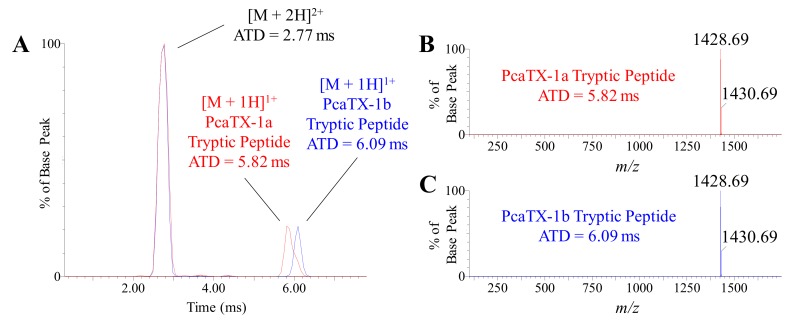
Representative ion mobilograms of PcaTX-1 tryptic peptides. (**A**) The EIMs of the doubly and singly charge states for PcaTX-1a (red) and PcaTX-1b (blue) tryptic peptides at *t*_R_ 4.21 min and 4.34 min (Figure 6A), respectively, revealed there is no significant difference in the ATDs for the doubly charge states, but showed a significant difference in the ATDs for the singly charge states. (**B**) and (**C**) show the ATD and corresponding mass spectra for the two conformers with an ATD of 5.82 and 6.09 ms, respectively. The CCS of the doubly charge states were both 195 Å^2^. However, the CCS of the digestion fragment from PcaTX-1a of 347 Å^2^ was significantly different from CCS of the PcaTX-1b of 356 Å^2^. The ion mobility was a definitive conformational analysis for the presence of the isomerized product of *Asp* in PcaTX-1. The red color used indicates the presence of *isoAsp* and the blue color used indicates the presence of *Asp*.

**Figure 10 toxins-12-00207-f010:**
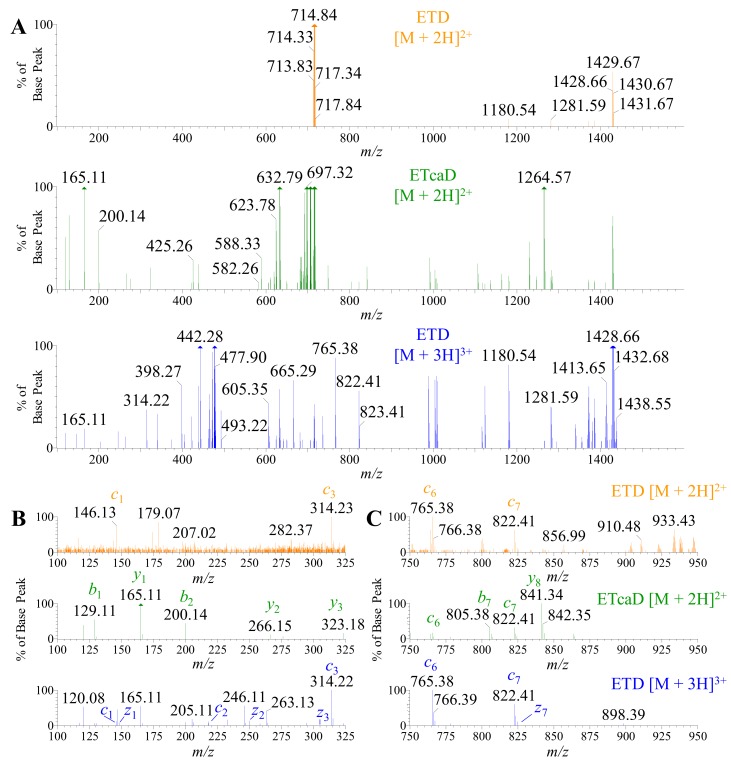
Comparison of ETD and ETcaD spectra of tryptic peptides for doubly and triply charged ions, respectively. (**A**) The ETD and ETcaD spectra of the doubly charged parent ion 714.84 *m/z* (orange—top and green—middle panel, respectively) was compared to the ETD spectra of the triply charged parent ion 476.88 *m/z* (blue—bottom panel) for PcaTX-1b tryptic peptide (Figure 6A: *t*_R_ = 4.34 min) for comparison. Oftentimes, trypsin digestion yields small fragment peptides with a high abundance of doubly charged ions, but not 3+ or higher charged states. ETD alone may not produce adequate fragmentation of the doubly charged state (orange—top panel). To enhance this fragmentation, an application of voltage to the transfer cell (~5–20 V) resulted in “secondary activation” and an increase in daughter ions (green—middle panel). This transfer cell voltage resulted in fragmentation that yielded more *c* and *z* ions, but also displayed a concomitant abundance of classic *b* and *y* ions. All three panels were magnified 75X the base peak to show low abundance fragmentation ions. (**B**) The magnified view of the ETcaD spectra at 100–325 *m/z* revealed the presence of *b* and *y* ions not seen in ETD experiments for either charged state fragmentation. The ETcaD spectra did not reveal any additional *c* or *z* ions as expected. (**C**) The magnified view of the ETcaD spectra at 750–975 *m/z* revealed the presence of *c* and *z* ions, but revealed a greater level of *b* and *y* ion formations. ETD on the doubly charged ion (green—middel panel of B and C) does not reveal the richness of data as seen in the triply charged state fragmentation (where complete coverage was achieved). Second, the application of secondary activation (ETcaD) forms “hybrid” spectra that do not greatly enhance the production of *c* and *z* ions but do augment the production of *b* and *y* ions. Taken together, the use of ETcaD may slow the de novo process without much benefit depending on the peptide under investigation and should be used with caution.

**Figure 11 toxins-12-00207-f011:**
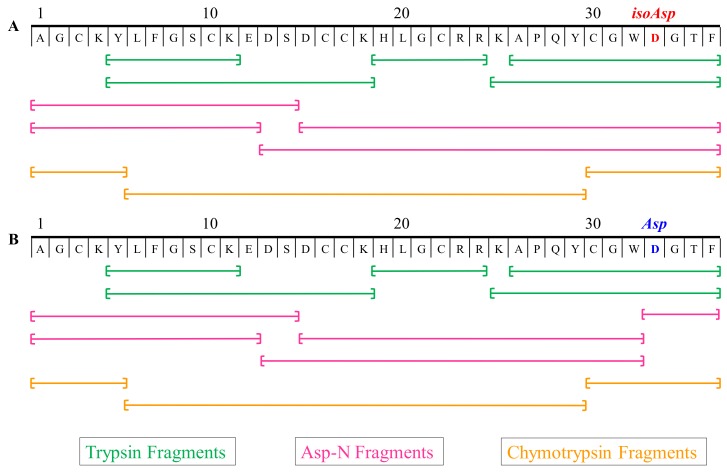
Digestion map of enzymatic peptides sequenced for PcaTX-1. (**A**) PcaTX-1a and (**B**) PcaTX-1b were digested with three different enzymes, i.e., trypsin, chymotrypsin, and Asp-N, that allowed for 100% coverage for de novo sequencing. All of the sequence fragments were positively identified with both CID and ETD, as discussed previsouly. All of the terminal fragments with *Phe* showed C-terminal amidation. The Asp-N fragments displayed and confirmed the expected *isoAsp* and *Asp* differences between PcaTX-1a and PcaTX-1b. The endoprotease Asp-N cleaves on the amino side of an *Asp* residue but not on an *isoAsp* residue; the N-terminal side cannot cleave the beta amino acid due to the presence of an extra methylene group in the peptide backbone. The use of multiple enzymes coupled to CID/ETD is a prudent practice for definitive identification.

**Table 1 toxins-12-00207-t001:** VGSC Na_v_1.7 modulatory properties of PcaTX-1 analogs.

Parameter	Control	PcaTX-1a	PcaTX-1b	*p* _ANOVA_
*I V*_½_ (mV)	−32.9 ± 0.889	−33.6 ± 0.683	−14.5± 0.592 ^*,†^	<0.000001
*I V*_rev_ (mV)	67.7 ± 0.667	70.6 ± 0.384	68.6 ± 1.27	0.08
*G V*_½_ (mV)	−28.6 ± 0.861	−28.4 ± 0.768	−6.34 ± 0.490 ^*,†^	<0.0000001
*G k*	3.34 ± 0.297	3.38 ± 0.482	5.47 ± 0.332 ^*,†^	0.006
*h* _∞_ *V_½_ (mV)*	−58.6 ± 1.18	−48.6 ± 1.16^*^	−54.1 ± 0.889	0.02
*h_∞_ k*	9.48 ± 0.176	9.47 ± 0.355	9.55 ± 0.367	0.1

*V_½_*, potential at which *I*_Na_ is half of the maximum; *V*_rev_, reversal potential; *G,* Na^+^ conductance; *h*_∞_ steady-state inactivation parameter; *V_½_* and *k*, mid-activation, or inactivation voltage and slope factor of the fit for the *G/V* and *h*_∞_ curves; * values are statistically different than control by a Student’s *t*-test (*p* < 0.05); † values are statistically different than PcaTX-1a by a Student’s *t*-test (*p* < 0.05); values are presented as mean ± SEM (*n* = 5).

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
