# Peer review of "Aspartic Acid Isomerization Characterized by High Definition Mass Spectrometry Significantly Alters the Bioactivity of a Novel Toxin from Poecilotheria"

_toxins, 2020, doi:10.3390/toxins12040207_

Round 1

Reviewer 1 Report

The article entitled “Aspartic acid isomerization characterized by high definition mass spectrometry significantly alters the bioactivity of a novel toxin from Poecilotheria” addresses an interesting approach using a combination of high-resolution MS-MS with ion mobility spectrometry (HDMS E). Authors demonstrated the suitability of HDMS E in toxins screening, allowing them to detect isobaric conformers originating from non-enzymatic Aspisomerization from the venom of ten “tiger” spider (Genus: Poecilotheria) species. Moreover, authors found that such Asp isomerization constitute the first report in a toxin that affected their activity. Besides, the strategy employed yielded a two-novel voltage-gated sodium channel (VGSC Nav1.7)toxins from the venom of Poecilotheria vittata, named as PcaTX-1a and PcaTX-1b. In summary, the work is well presented: abstract, aims, introduction, methods, results, discussion, and conclusions are coherent. Results and Figures are clearly presented. As authors said, this study establishes the value of the HDMS E , and shows strong methodology, thus, must be considered by toxinologists in toxins biodiscovery. This works shows novelty and its finding could have some implication in toxins characterization, mainly in bio-guided assays.

Author Response

Dear Reviewer 1,
Thank you for reviewing our manuscript and thank you for your summary comments.
I did not see a request for any edits or changes and so have left the manuscript as is.

Reviewer 2 Report

Aspartic acid isomerization characterized by high definition mass spectrometry significantly alters the bioactivity of a novel toxin from Poecilotheria

In the present study, the authors created a high definition approach that fuses conventional high-resolution MS-MS with ion mobility spectrometry (HDMSE). They investigated venom from ten species of “tiger” spider (Genus: Poecilotheria) and discovered they contain isobaric conformers originating from non-enzymatic Asp isomerization. As it known, structural characterization and identification of post-translational modifications are essential to develop biological lead structures into pharmaceuticals. In my opinion, the study is interesting and innovative, including was well delineated. However, I have some comments:

Comment (1): The text is a little bit confusing and needs a thorough revision of English.

Comment (2): In my opinion, title is good but it should be changed to be more attractive.

Comment (3): Abstract. The background topic is poor. It is necessary include at least one more clear objective for study.

Comment (4): Introduction. There is a brief review of existing knowledge and relevance of study. I suggest change in the following sentences:

- Line 103-104: There is a mistake in the arrangement of the disulfide bridge of the poecilotheriatoxin (PcaTX-1).

- I suggest to the authors to add a short paragraph about the biological activities of the animal toxins. Moreover, how modifications in their structures can effect these biological activities.

- General comment: Proofreading is strongly recommended to ensure that the text is easily understandable. In addition, the introduction lacks a lot of information about the Poecilotheria venom components, as it is the main subject of the manuscript. More background should be addressed in the introduction.

Comment (5): Results. Very good written results. A lot of information are included.

 Comment (6): Discussion. Great information in this section but need a proofreading to improve it well.

Comment (7): Conclusion. The authors concluded their works at the end systematically.

Comment (8): Materials and Methods. It is clear and the details are sufficient to understand the strategies adopted for the development of the study.

Comment (9): Figures and legends. Legends are adequate and figures are necessary to understand the results obtained.

Comment (10): References. Although, there is a big list of references (93 references) in comparison to the manuscript size, there are some references do not require in this manuscript.

Reviewer 3 Report

The manuscript under reviewing provides different procedures for screening and characterization of venom compounds from ten species of “tiger” spider. The authors made a “big job” - I think the manuscript is very interesting and well written and I do not have any comments either on the experiments or on the results or the discussions. In general, the manuscript looks laconic and is suitable for publication in Toxins.  

Author Response

Dear Reviewer 3,
Thank you for reviewing our manuscript and thank you for your summary comments.
I did not see a request for any edits or changes and so have left the manuscript as is.